# Evaluation of EEG Oscillatory Patterns and Classification of Compound Limb Tactile Imagery

**DOI:** 10.3390/brainsci13040656

**Published:** 2023-04-13

**Authors:** Kishor Lakshminarayanan, Rakshit Shah, Sohail R. Daulat, Viashen Moodley, Yifei Yao, Puja Sengupta, Vadivelan Ramu, Deepa Madathil

**Affiliations:** 1Neuro-Rehabilitation Lab, Department of Sensors and Biomedical Engineering, School of Electronics Engineering, Vellore Institute of Technology, Vellore 632014, Tamil Nadu, India; 2Department of Chemical and Biomedical Engineering, Cleveland State University, Cleveland, OH 44115, USA; shahrakshit23@gmail.com; 3Department of Physiology, University of Arizona College of Medicine, Tucson, AZ 85724, USA; 4Arizona Center for Hand to Shoulder Surgery, Phoenix, AZ 85004, USA; 5Soft Tissue Biomechanics Laboratory, Med-X Research Institute, School of Biomedical Engineering, Shanghai Jiao Tong University, Shanghai 200030, China; 6Jindal Institute of Behavioral Sciences, O. P. Jindal Global University, Sonipat 131001, Haryana, India

**Keywords:** motor imagery, tactile imagery, brain–computer interface, event-related desynchronization

## Abstract

**Objective**: The purpose of this study was to investigate the cortical activity and digit classification performance during tactile imagery (TI) of a vibratory stimulus at the index, middle, and thumb digits within the left hand in healthy individuals. Furthermore, the cortical activities and classification performance of the compound TI were compared with similar compound motor imagery (MI) with the same digits as TI in the same subjects. **Methods:** Twelve healthy right-handed adults with no history of upper limb injury, musculoskeletal condition, or neurological disorder participated in the study. The study evaluated the event-related desynchronization (ERD) response and brain–computer interface (BCI) classification performance on discriminating between the digits in the left-hand during the imagery of vibrotactile stimuli to either the index, middle, or thumb finger pads for TI and while performing a motor activity with the same digits for MI. A supervised machine learning technique was applied to discriminate between the digits within the same given limb for both imagery conditions. **Results:** Both TI and MI exhibited similar patterns of ERD in the alpha and beta bands at the index, middle, and thumb digits within the left hand. While TI had significantly lower ERD for all three digits in both bands, the classification performance of TI-based BCI (77.74 ± 6.98%) was found to be similar to the MI-based BCI (78.36 ± 5.38%). **Conclusions:** The results of this study suggest that compound tactile imagery can be a viable alternative to MI for BCI classification. The study contributes to the growing body of evidence supporting the use of TI in BCI applications, and future research can build on this work to explore the potential of TI-based BCI for motor rehabilitation and the control of external devices.

## 1. Introduction

Motor imagery (MI) is the mental practice of a specific motor activity without the concomitant physical movement [1]. MI activates the same areas of the brain as actually performing the action such as primary motor cortex, premotor cortex, and sensorimotor cortex [2,3,4]. Furthermore, MI has been shown to induce neuroplasticity, resulting in improvements in motor learning in athletes [5], musicians [6], healthy individuals [7,8,9], and patients [10]. One mechanism through which MI affects motor performance is by eliciting an alpha band (8–12 Hz) and beta band (13–30 Hz) event-related desynchronization/synchronization (ERD/ERS) during imagination of the movement [11,12,13]. The alpha ERD is thought to reflect the inhibition of task-irrelevant sensory input and the enhancement of attentional focus, while the beta ERD is thought to reflect the activation of the motor cortex and the planning and execution of movement. The ERD/ERS enables the use of MI in brain–computer interface (BCI) applications such as robotic prostheses control [14], wheelchair navigation [15] as well as a training regime to improve the motor performance in any individual. MI-based BCI offers a better mapping between the control command and the interface being controlled compared to other existing BCI modalities such as steady state visually evoked potential [16] and event-related potential based BCI systems [17]. Unlike the other BCI systems, MI does not require external stimuli [18,19].

Imagining a tactile sensation has been reported to elicit similar cortical excitations to MI in the form of ERD/ERS [20]. Unlike MI, which simulates physical movement through imagery of a body part, tactile imagery (TI) involves focusing on the somatosensory experience of a particular body part. Such TI has been shown to activate the primary somatosensory cortex [21], while MI activates the motor cortex. Consequently, TI could be a unique mental strategy to help rehabilitate people who have limited sensory abilities. TI-based BCI classification has shown promise in improving BCI performance. A recent study on 106 healthy subjects investigated the accuracy of TI-based BCI discrimination and found 70.7% of subjects achieved a performance above 70% [22], suggesting TI as a new modality for BCI development. However, studies on TI-based BCI are few and have largely evaluated left- and right-hand discrimination [22].

Although conventional simple imagery involving left and right side discrimination may exhibit higher BCI accuracy due to the clear neurophysiological differences between the two sides of the body [23]); this simple discrimination has limited use for individuals with paralytic stroke on one side of the body, since the neurofeedback from the paralytic side is more important for rehabilitation [24,25]. Furthermore, due to the restricted number of available classes, the BCI output commands are also limited in simple imagery. For instance, the use of both hand movement imagination was implemented to facilitate continuous three-dimensional control of a virtual helicopter in a three-dimensional space as a means of compensating for the shortage of instructions in a simple limb motor imagery-based BCI [26]. To address such asymmetrical cortical activations due to motor deficits and limited output commands, compound imagery involving several parts of a single given limb has been proposed to replace simple imagery [27]. Compound imagery has advantages in BCI applications by expanding the number of output commands available from a single limb [28]. However, very few studies have reported on the brain oscillatory patterns during compound movement imagery. Furthermore, no study has reported on compound tactile imagery to our knowledge. A compound TI involving digits in the same given limb would increase the number of BCI commands available since compound imagery can lead to the activation of neuron oscillation in several functional areas of the cerebral cortex. Additionally, compound TI has the potential to be incorporated with compound MI, even furthering its benefits, and can play an important role in the rehabilitation of upper extremity.

Therefore, the purpose of this study was to investigate the effect of a compound TI on the cortical oscillatory activity during imagery of a vibratory stimulus at the index, middle, and thumb digits within the left hand in healthy individuals. TI-induced cortical activities and BCI digit classification performance was compared to a similar compound MI involving imagining finger movements with the same digits as TI on the left hand in the same subjects. To achieve this, we examined the brain activities with the help of an electroencephalogram (EEG) during both TI and MI. A supervised machine learning technique was applied to discriminate between the digits within the same given limb. It was hypothesized that the compound TI would show similar oscillatory patterns and perform similarly to its MI counterpart in the digit discrimination.

## 2. Methods

### 2.1. Subjects

Twelve healthy adults (seven males and five females) with a mean age of 27.18 ± 5.79 years volunteered in the study. The number of subjects was chosen based on recent EEG studies with similar small sample sizes [29,30,31]. All subjects had no history of upper limb injury, musculoskeletal condition, or neurological disorder. None of the subjects had any prior experience with motor or tactile imagery. The institutional review board of the Vellore Institute of Technology approved the protocol. Subjects read and signed a consent form before participating in the experiment.

### 2.2. Procedure

The ERD and BCI classification performance on discriminating between the digits in the left hand during tactile imagery of a vibratory stimuli to either the index, middle, or thumb finger pads was evaluated and compared to a MI task involving imagining a button push using the same digits as TI.

### 2.3. EEG Recording

EEG signals were recorded using the Allengers’ VIRGO-32 EEG system (Allengers Medical Systems, Chandigarh, India) in accordance with the international 10–20 system. The VIRGO-32 system consisted of a 20-electrode EEG cap placed on the subject’s scalp, with electrodes at FP1, FPz, FP2, F7, F3, Fz, F4, F8, T3, C3, Cz, C4, T4, T5, P3, Pz, P4, T6, O1, and O2. The reference electrode was placed at Fz, and the ground electrode was placed at FPz. Prior to the electrode placement, the skin was cleaned, and a conductive gel was applied to ensure that the impedance was below 5 kΩ.

### 2.4. Equipment

A sensory-motor box (Figure 1a) was constructed to aid the subjects in their TI and MI training. The box consisted of three push buttons with small flat vibration micro motors (Sunrobotics, Gujarat, India) attached on top of the buttons. Adjustable sliders were integrated into the box to align each button with the corresponding digit pads. The vibration motor produced a 27-Hz sine wave, which fell within the frequency range to stimulate the Pacinian and Meissner corpuscles [32]. The sensory-motor box was controlled by an Arduino UNO R3 microcontroller (Arduino, Ivrea, Italy).

### 2.5. Experimental Design

To provide a visual cue to let the subjects know when to perform the imagery and offer action observation, a 3D left hand was modeled and animated in Blender software (Blender Foundation, Amsterdam, The Netherlands) to perform finger movement tasks using the index, middle, and thumb digits. A 3D virtual environment was built using the Unity game engine (Unity Technologies, San Francisco, CA, USA) where the hand and its animation was gamified. A computer monitor was placed in front of the subject in which the 3D hand animations and cues to perform the imagery were displayed.

The experiment was conducted in a quiet room with minimal distractions. Subjects were comfortably seated with their arms placed on the armrests and were instructed to avoid any movement including eye blinking to reduce motion artifacts during EEG recording. The experiment comprised two imagery conditions, namely, TI and MI, with each condition consisting of one block per digit (index, middle, and thumb), making a total of six blocks (2 imagery conditions × 3 digits).

Each block of TI consisted of five sessions, with each session beginning with five training trials followed by 10 imagery trials, making it a total of 15 trials per session. For the training trials, a brief vibrotactile stimulation lasting 150 ms was provided at the start of the trial to either the index, middle, or thumb digit pad, depending on the block. The imagery trials began with a text cue displayed over the 3D hand, instructing the subjects to imagine the same vibrotactile stimulation on the corresponding digit pad. After the text cue disappeared, the subject imagined the vibratory stimulation for a brief duration of 150 ms. Each trial had a 4-s rest period following the vibrotactile stimulation and its imagination (Figure 1b). A text reminded the subject of the digit being tested for 3 s before the start of each block.

A single block of MI consisted of five sessions of 10 trials per session, where a text reminded the subject of the digit being tested for 3 s before the block started. At the start of the trial, the 3D animation showed the digit pressing down on a button, holding the position for 2 s, releasing the button, and returning to the initial position followed by a 4-s rest (Figure 1c). Subjects were asked to observe the task animation and imagine the same movement kinesthetically by forming an impression of their own left-hand digits performing the task. Prior to the MI tasks with each digit, the subjects performed a trial run by physically performing the button-pushing task with the corresponding digit on the sensory-motor box to familiarize themselves with imagining the task at the pace of the 3D animation.

The order of the imagery conditions and the order of the digits within each imagery condition was randomized for each subject. EEG signals were continuously recorded at a sampling rate of 250 Hz throughout the experiment.

### 2.6. EEG Analysis

MATLAB (R2022a, The MathWorks, Natick, MA, USA) was used for the ERD and digit-discrimination analysis. The EEGLAB toolbox was utilized for ERD analysis, and an artificial neural network from the Neural Network toolbox in MATLAB was used for the discrimination analysis. The EEG signals were filtered using a bandpass filter between 0.5 and 50 Hz to remove the line noise and re-referenced to a common average reference, where the average of the signal at all the electrodes was computed and subtracted from the EEG at each electrode at each time point. The ADJUST algorithm [33] is an automatic algorithm designed to identify and remove artifacts from EEG data. It uses ICA to isolate components with artifacts in EEG recordings. ADJUST detects ICA components that contain artifacts by merging features that capture common artifact-specific spatial and temporal characteristics such as blinks, eye movements, and other discontinuities. After identifying the independent components with artifacts, this algorithm eliminates them from the data, leaving the neural source activity almost intact. The cleaned data were segmented into epochs spanning from −1000 ms to 3000 ms relative to the start of each trial for both the TI and MI blocks.

### 2.7. ERD Extraction

To investigate brain activity during the imagery tasks, nine electrodes, namely, F3, Fz, F4, C3, Cz, C4, P3, Pz, and P4, were chosen since they encompass the frontal, central, and parietal regions of the scalp, respectively. The electrodes were chosen to encompass the entire sensory and motor area and to account for any of the expected contralateral activation as well as any possible ipsilateral activation. The EEG data were analyzed using time–frequency analysis to generate event-related spectral perturbations (ERSPs). ERSPs reflect the fluctuations in frequency power over time. This was achieved by dividing the mean event-related spectrum at each time–frequency point by the mean spectral estimate obtained during the pre-stimulus baseline period at the same frequency as follows:(1)$$\mathrm{ERSPlog}(f,t)=10\log_{10}\left[\frac{\mathrm{ERS}\left(f,t\right)}{\mu_{B}\left(f\right)}\right]$$
where ERS is the mean event-related spectrum computed using the formula
(2)$$\mathrm{ERS}\left(f,t\right)=\frac{1}{n}\sum_{k=1}^{n}{\left|\mathrm{Fk}\left(f,t\right)\right|}^{2}$$
where n is the total number of trials, and Fk(f, t) is the spectral estimate at frequency f and time point t for trial k.

μ_B_(f) is the mean spectral estimate for all baseline points at frequency f given by the formula
(3)$$\mu_{B}\left(f\right)=\frac{1}{\mathrm{nm}}\sum_{k=1}^{n}\sum_{\;t^{\prime }\in B}^{\;}\;{\left|\mathrm{Fk}\left(f,\;t^{\prime }\right)\right|}^{2}$$
where B is the ensemble of time points in the baseline period and m is the cardinal of B or the total number of time points in the baseline period.

The epochs were uniformly divided into 200 time points, and the ERSP values for each epoch were normalized to its baseline spectra. Then, the averaged ERSP values of all the task epochs were calculated. The average alpha and beta band ERSP were calculated by averaging the amplitude values within the alpha (8–12 Hz) and beta (13–30 Hz) frequency ranges, respectively. ERD was seen as a decrease in ERSP power during the performance of the imagery task. The time–frequency analysis utilized a decibel (dB) scale to provide a more comprehensible display of power alterations over time, since it represents power variations in relation to the baseline. Consequently, the dB scale allows for an enhanced perception of frequency band fluctuations compared to just examining the raw power spectrum.

### 2.8. Discriminant Analysis

To evaluate neural activity discrimination of three digits, an artificial neural network (ANN) was constructed using the nrptool module in MATLAB. The toolbox employed a two-layer feedforward network with a learning procedure based on the scaled conjugate gradient backpropagation algorithm. By integrating a scaling procedure that dynamically determines the direction of the gradient during optimization, the algorithm unites the conjugate gradient (CG) approach. This integration can result in quicker convergence and superior precision when compared to alternative optimization techniques such as gradient descent or Gauss–Newton.

Feature extraction was performed on the ERD data to extract discriminating features associated with the three digits during the imagery tasks. For both TI and MI, a 1-s time period between 0 ms and 1000 ms in the epoch where the ERD occurred was extracted from the ERSP data in each trial within the alpha-band frequency (8–12 Hz) and beta-band frequency (13–30 Hz) for each of the nine electrodes (F3, Fz, F4, C3, Cz, C4, P3, Pz, and P4). The extracted ERD data for each frequency band (alpha and beta) were concatenated and fed into the ANN, which had an input layer with 150 neurons (N) corresponding to the 50 imagery trials per digit after removing the vibratory training trials, an output layer with three neurons (M) corresponding to the three digit classes, and a hidden layer with 22 neurons calculated using the formula $\sqrt{N\times M}$ [34].

To train the ANN, a “training set” of 70% of the data was randomly selected. A total of 15% of the remaining data was held back and used as “validation data” to validate the model, and the remaining 15% was held back as “testing data” to evaluate the model. The neural network training was repeated 100 times to minimize the influence of random fluctuations from the training set during each iteration. The accuracy rates from the 100 runs were averaged to obtain the final accuracy rate.

### 2.9. Statistical Analysis

To compare the ERD elicited during TI vs. MI, a two-way repeated measures ANOVA was performed on the digit task-related average ERD for the alpha and beta band separately. Specifically, the alpha and beta bands of the ERD were ERSP averaged over the 1-s imagery period immediately after the rest cue for each of the two frequency bands. The ERD was calculated from the ERSP at the C4 electrode. C4 was chosen since they are placed over the contralateral sensorimotor area. The analysis included the imagery condition (TI vs. MI) and digit (index, middle, and thumb) as independent variables. Pairwise comparisons were performed using post hoc Bonferroni *t*-tests. Additionally, a paired *t*-test was used to compare the classification accuracy between the two imagery conditions. The statistical analysis was conducted with SigmaStat 4.0 (Systat Software Inc, San Jose, CA, USA), and the statistical significance was set at α = 0.05.

## 3. Results

### 3.1. Time–Frequency Analysis of EEG

The present study involved conducting trials to investigate tactile and motor imagery using digits on the left hand. The experiment measured brain activity through EEG in order to study the activation of the sensorimotor area during these tasks. Figure 2 illustrates the grand average time–frequency maps of all subjects at C4, for all three hand digits under both imagery conditions. The time–frequency maps demonstrate ERD as a decrease in power in the alpha and beta frequency bands during imagery. For TI, a 150 ms vibratory stimulus was imagined at 0 s, resulting in a band power decrease that was consistent across all trials in C4. The maximum cortical activity in the form of ERD was observed at around 500 ms after the vibrotactile sensory imagination for all three digits. Similarly in MI, a long-lasting ERD was seen at the onset of the digit task and lasted for the 2 s duration the task was imagined.

Moreover, to further understand the activation of the sensorimotor area, the average energy distribution of the combined alpha and beta frequency bands (8–30 Hz) of each channel was calculated and plotted into a topology map based on the channel positions (Figure 3). The results showed that the mean EEG potential during both the tactile and motor imagery of the left-hand digit tasks was topographically focused in the contralateral right sensorimotor area, consistent with previous studies [35].

A repeated-measures ANOVA was applied to study the differences between TI vs. MI. The assumptions of the ANOVA on the ERD outcomes were first verified. No significant outliers were detected in the data, and the results were normally distributed with no violations of normality observed in any group with a *p* > 0.05, as indicated by a Shapiro–Wilk normality test. Additionally, the Brown–Forsythe test was conducted to assess whether the variances of the differences between the vibration conditions were equivalent, with the results indicating equal variances for all groups (*p* > 0.05).

In the alpha band, a repeated measures ANOVA showed that ERD significantly differed by the main effect of imagery condition (TI vs. MI) (*p* = 0.014), but not by digit (index, middle, and thumb) (*p* = 0.647), or their interactions (*p* = 0.771). Post hoc Bonferroni tests demonstrated that ERD (Figure 4) was significantly higher in MI than TI for the index (*p* = 0.023), middle (*p* = 0.041), or thumb (*p* = 0.011) digits.

In the beta band, the repeated measures ANOVA showed that ERD significantly differed between the imagery conditions (*p* = 0.001), but not by digit (*p* = 0.613). The interactions between the imagery conditions and digits were not significant (*p* = 0.412). As shown by the post hoc Bonferroni tests, the ERD was significantly higher in MI compared to TI (Figure 4) for the index (*p* = 0.001), middle (*p* = 0.027), and thumb (*p* = 0.019) digits.

### 3.2. BCI Classification Analysis

In addition to the ERD analysis, a digit discrimination analysis was performed using a neural network for both tactile and motor imagery conditions. The results of the classification accuracy percentage for both the tactile and motor imagery for each subject is presented in Figure 5. To evaluate the statistical difference between the classification accuracies from the two imagery conditions, a paired *t*-test was conducted. The results showed that there was no significant difference (*p* = 0.821) between the digit classification accuracy in the tactile imagery (mean ± SD = 77.74 ± 7.89%) and motor imagery (mean ± SD = 78.36 ± 6.51%). The percentage accuracy and the mean absolute percentage error are shown in Table 1.

## 4. Discussion

The current study investigated the effect of tactile imagery on cortical oscillatory activity during tactile imagery of a vibratory stimulus at the index, middle, and thumb digits within the left hand in healthy individuals. The cortical activities and BCI performance from the compound tactile imagery involving digits from the same hand were compared to compound motor imagery involving imagining finger movements with the same digits as TI in the same subjects. Both TI and MI induced a clear oscillatory power decrease in the alpha and beta bands in the sensorimotor areas, as seen from the EEG data. The presence of an identifiable pattern of ERD, seen as an oscillatory power decrease in the alpha and beta bands, is an indicator of the performance of imagery [11]. Furthermore, the study also presented a novel TI-based BCI involving the classification of digits from a single limb using ERD induced during TI as features and compared it to MI. To the best of our knowledge, this is the first study to propose and validate such a BCI using compound tactile imagery. The three-class compound tactile imagery BCI in the current study showed an average accuracy of 77.74 ± 6.98%, which were comparable to the conventional two-class TI-based BCI for left vs. right hand discrimination by Yao et al. [22], which demonstrated a mean classification accuracy of 78.9 ± 13.2%. Such compound TI shows promise in developing a novel paradigm where TI can be combined with MI-based BCI to expand the number of BCI inputs.

Tactile imagery of a short vibratory burst in the left-hand digits induced an EEG power decrease in the form of ERD in both the alpha (8–12 Hz) and beta (13–30 Hz) bands in the contralateral right hemisphere, focused on the C4 electrode over the sensorimotor cortex. The ERD induced during TI were in accordance with other studies that have shown similar ERD with vibratory stimulus applied to the hand [19,36]. The results from the current study suggest that tactile imagery produces a similar ERD to experiencing an actual vibratory stimulus. Such TI of a short vibratory stimulus might lead to improvements in performing motor tasks, as seen by our previous study where subjects showed an improvement in reaction time following TI training [37]. A possible explanation could be that the tactile imagery has an effect on the online representation of the hand in space for the subjects, as previous studies have shown that imperceptible vibration enhances tactile sensation [38,39,40], and that tactile stimulation improves proprioception of the hand [41,42]. The ERD effect observed during TI was spread over the primary motor cortex and the somatosensory area (Figure 3), which suggests that the sensory stimulation was received at the somatosensory area and processed further in the primary motor cortex. This is consistent with previous research showing that sensory feedback influences the discharge of corticospinal cells in the motor cortex [43,44].

Although the ERD induced by TI was significantly lower than MI for all three digits, the TI-induced ERD still showed a high discriminative spatial pattern between the three digits from the same hand. Previous studies using BCI based on steady-state somatosensory evoked potentials (SSSEP) to repeated tactile stimuli reported that over 80% of the subjects showed an accuracy below 70% [45], which may limit the potential of tactile BCIs. Another similar study using SSSEP-based BCI showed an average performance of 58% over a group of 16 subjects [46]. In contrast, the current study demonstrated that TI-induced ERD dynamics can lead to a novel TI-based BCI with a higher performance of around 78%, with the majority of subjects showing a classification above 70%. The activation of the sensorimotor cortex ERD by unilateral tactile imagery provides high spatial discrimination. These findings suggest that the TI-induced ERD approach can be useful in reducing the calibration effort in multi-class compound BCI systems, and has implications in BCI-based prosthetics and robots. Future studies could focus on improving the spatial resolution such as by combining TI and MI.

In the current study’s experimental setup, both action observation (AO) and MI were used. It has been demonstrated that AO + MI causes a greater desynchronization [47]. Clear instructions were given to the subjects to visualize the observed action kinesthetically in order to ensure that they completed both the AO and MI simultaneously. It has been suggested that AO and MI training should be combined and used simultaneously and should not be perceived as mutually exclusive forms of treatment based on a number of studies that were examined by Vogt et al. [48].

The current study proposes a methodology that was developed based on previous work on MI and TI, which introduced subthreshold vibration as a method to improve BCI performance [36], and used TI training to improve the reaction time [37]. In this study, the proposed approach of imagining a tactile sensation in the digits of a signal given limb was shown to induce a cortical response and BCI performance similar to MI. The potential applications of this methodology include stroke patients with motor impairment but preserved sensory abilities, who may benefit from the current approach in improving their rehabilitation regime.

## 5. Limitations

The major limitation of this study was the presence of event-related potential (ERP) components from the action observation that may have interacted with the ERD elicited by the kinesthetic motor imagery. The experimental design in the current study made it impossible to separate the AO related ERP components from the MI related ERD. Future studies could run practice sessions with an AO component to train the subjects on the pacing and timing of the performed kinesthetic MI, and for the actual experiment, the AO was turned off and the subjects were made to perform MI without a visual cue. While the oscillatory dynamics induced by TI were comparable to those induced by MI, they might have different cortical activation sources. Future studies should employ functional near-infrared spectroscopy (fNIRS) combined with EEG to offer better spatial resolution and help increase the understanding of TI and MI. The study mainly consisted of young adult subjects, and future studies should aim to broaden the age range to evaluate the age effect. Additionally, the current subjects did not undergo extensive BCI training, which has been demonstrated to enhance BCI performance. A longer duration study involving multiple sessions of TI would be necessary to evaluate the effect of TI-based BCI training on improving BCI-illiteracy. Finally, due to the limited sample size and observed variations in individual subject performances, caution should be exercised when interpreting the results of this study, as these factors may have impacted the statistical power.

## 6. Conclusions

In summary, the current study evaluated the effect of a novel compound tactile imagery of a vibratory stimulus on the digits from a single given limb on the cortical response and BCI discrimination performance. The proposed compound tactile imagery showed a reduced ERD response but a similar BCI performance to a similar compound MI. The current approach also provides a novel paradigm to increase the number of BCI commands.

## Figures and Tables

**Figure 1 brainsci-13-00656-f001:**
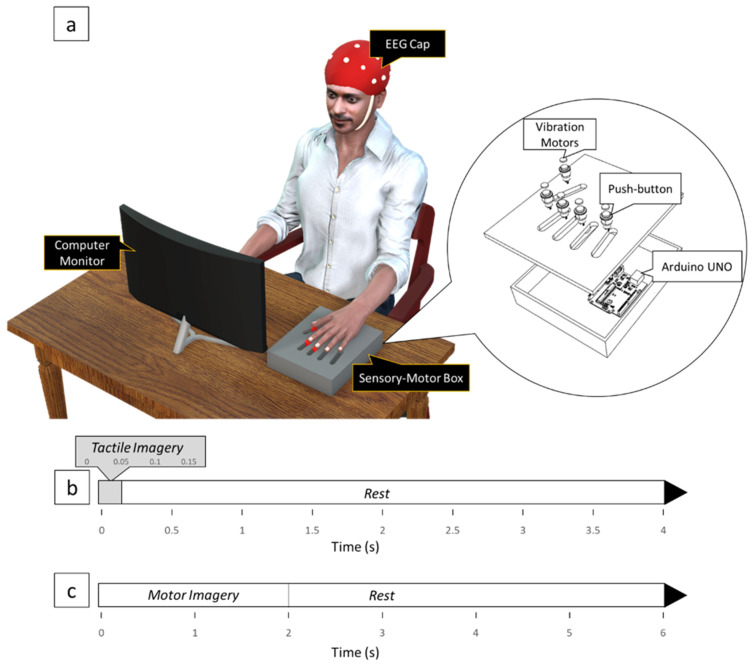
(**a**) Experimental setup. (**b**) Timeline of events for a single trial of TI. The vibratory stimulus was imagined for 150 ms at the beginning of each trial at either the index, middle, or thumb finger pad. (**c**) Timeline of events for a single trial of MI. Subjects imagined a button-pushing task using either the index, middle, or thumb digits.

**Figure 2 brainsci-13-00656-f002:**
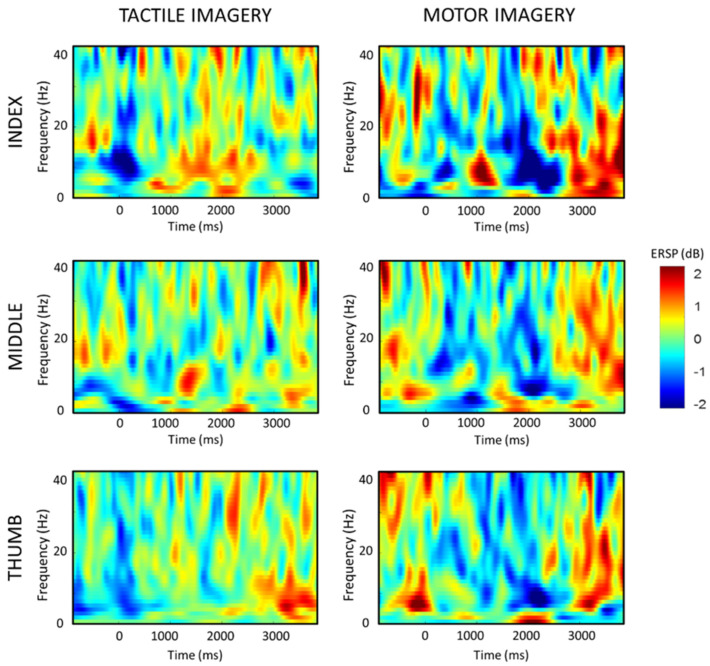
Averaged time–frequency maps of all participants. Blue indicates ERD.

**Figure 3 brainsci-13-00656-f003:**
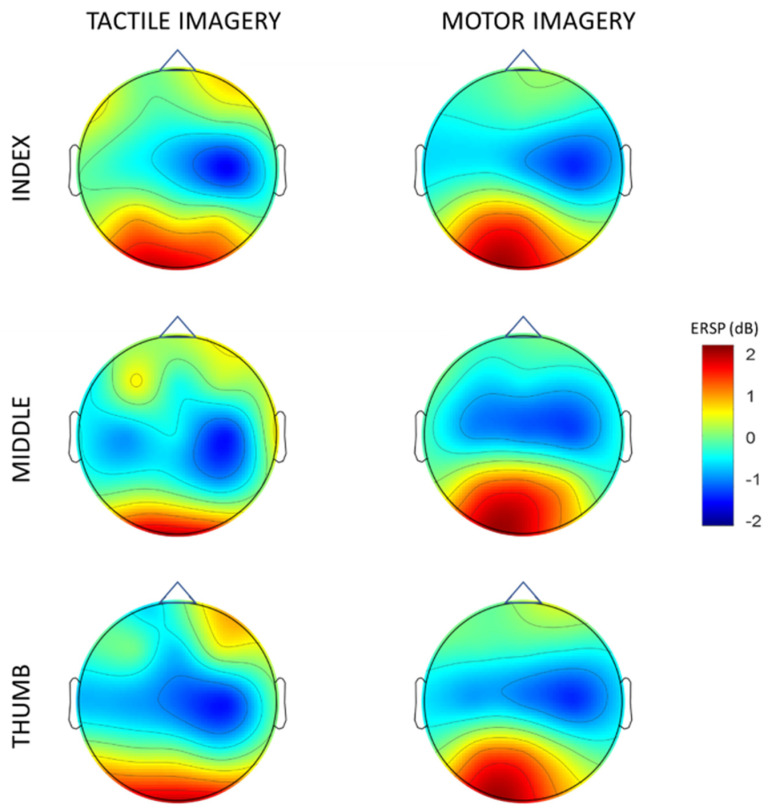
Averaged topographical distribution of power. Blue indicates ERD.

**Figure 4 brainsci-13-00656-f004:**
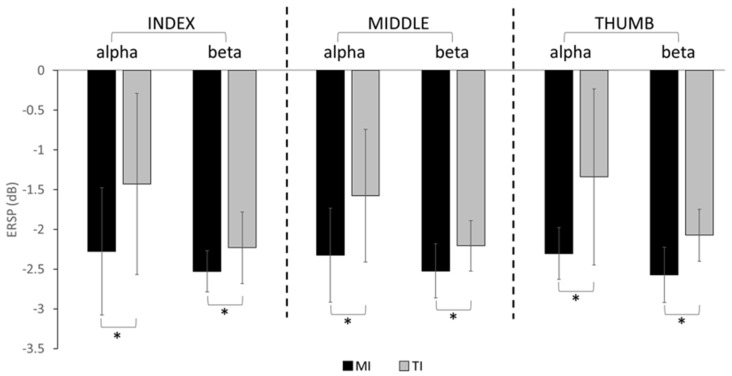
The grand average ERD (mean + SD) of all participants from the alpha and beta bands for both imagery conditions for each digit. A significant difference between the imagery conditions is noted with *.

**Figure 5 brainsci-13-00656-f005:**
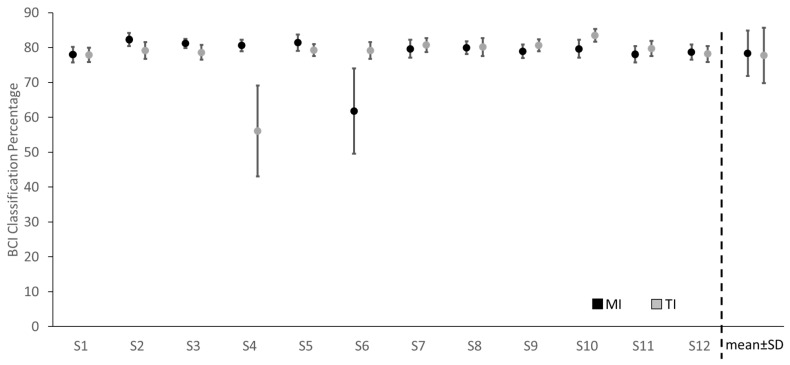
Classification percentage for each subject and the mean classification percentage.

**Table 1 brainsci-13-00656-t001:** Mean absolute percentage error (MAPE) and prediction accuracy (PA) of the artificial neural network (ANN) model.

	MI		TI	
Subject	PA	MAPE	PA	MAPE
S1	78.00	22.00	77.90	22.10
S2	82.27	17.73	79.13	20.87
S3	81.20	18.80	78.67	21.33
S4	80.60	19.40	56.07	43.93
S5	81.40	18.60	79.27	20.73
S6	61.80	38.20	79.17	20.83
S7	79.70	20.30	80.70	19.30
S8	79.97	20.03	80.10	19.90
S9	78.93	21.07	80.63	19.37
S10	79.67	20.33	83.50	16.50
S11	78.13	21.87	79.73	20.27
S12	78.70	21.30	78.10	21.90

## Data Availability

The datasets used and/or analyzed during the current study are available from the corresponding author on reasonable request.

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
