# Peer review of "Evaluation of EEG Oscillatory Patterns and Classification of Compound Limb Tactile Imagery"

_brainsci, 2023, doi:10.3390/brainsci13040656_

Round 1

Reviewer 1 Report

Lakshminarayanan et al. evaluated event-related desynchronization (ERD) response and brain-computer interface (BCI) classification performance on discriminating digits for tactile imaging (TI) and motor imagery (MI). The authors have utilized a feedforward neural network model for classification purposes. Here are some of my comments:    Why only alpha and beta bands were utilized? What about the other spectral bands? The authors should comment and report the findings of all the measured EEG signals from participants and not only two selected bands. Please elaborate and modify the results accordingly.    Average frequency maps for participants is not a clear indication of the differences reported by the authors. The authors should indicate regions of statistical significance from both conditions. Alternatively, topographic maps of p-values resulting from the statistical test judged by a significance level of 0.05 between TI and MI computed in alpha and beta bands should be shown here.   The authors have pooled all the electrodes across participants together in one group. I don't think this is the correct approach. Do all the electrodes contribute to the same outcome? Is there no effect on the contribution of each hemisphere towards the task? No contribution of gender? Please elaborate.    Fig. 5 should show all the data points and not just the box plot.    The details of input variables for the neural network are not provided as well. Please provide a detailed layout of the neural network with variables and other details for the ease of understanding the model reported here. For the classification, the details of the neural network have not been reported here. The metrics for classification such as Logarithmic Loss, Confusion Matrix, Area under Curve, F1 Score, Mean Absolute Error, and Mean Squared Error, have not been reported. Furthermore, details of the training and testing groups have not been elaborated. Without assessing the performance of training and testing steps, it is difficult to accept the accuracy reports provided here. Kindly elaborate and update the results. 

Author Response

We thank the reviewer for the insightful comments and giving us the opportunity to improve our paper. We have addressed the comments as follows:

  1. Why only alpha and beta bands were utilized? What about the other spectral bands? The authors should comment and report the findings of all the measured EEG signals from participants and not only two selected bands.

We have added more information to justify the utilization of only alpha and beta band since the sensory and motor related event-related desynchronization activity shows up predominantly in the alpha and beta band respectively in the introduction section line 42-47.

  1. Average frequency maps for participants is not a clear indication of the differences reported by the authors. The authors should indicate regions of statistical significance from both conditions. Alternatively, topographic maps of p-values resulting from the statistical test judged by a significance level of 0.05 between TI and MI computed in alpha and beta bands should be shown here.  

We understand the reviewer’s comments on showing the time-frequency plot of the statistical significant differences between tactile imagery and motor imagery. However, the aim of the study was not to see if the conditions are significantly different but rather how similar they are in eliciting EEG signals that could be used for machine learning algorithms. We believe individual plots offers a better picture of the ERD elicited by the two conditions. Having said that, if the reviewer still feels strongly about the plot we will be happy to oblige and replace the figure.

  1. The authors have pooled all the electrodes across participants together in one group. I don't think this is the correct approach. Do all the electrodes contribute to the same outcome? Is there no effect on the contribution of each hemisphere towards the task? No contribution of gender? Please elaborate.   

The reviewer is correct in assuming that all the electrodes we have chosen would not contribute equally in terms of ERD. We have chosen 9 electrodes that covers the entire sensory and motor region on both sides of the scalp. This was done to include all expected contralateral activations as well as any potential ipsilateral activation too. The 9 electrodes were chosen to provide EEG signals to train our algorithm as such we did not want to create bias by cherry picking features such as electrodes from one particular hemisphere and so included the nine electrodes. However for the statistical analyses we only included data from C4 electrode on the contralateral somatosensory area. We have edited the methods section to add more clarity on this as per the reviewer's comments.

  1. Fig. 5 should show all the data points and not just the box plot.

We have remade the figure as per the reviewer's comments

  1. The details of input variables for the neural network are not provided as well. Please provide a detailed layout of the neural network with variables and other details for the ease of understanding the model reported here. For the classification, the details of the neural network have not been reported here. The metrics for classification such as Logarithmic Loss, Confusion Matrix, Area under Curve, F1 Score, Mean Absolute Error, and Mean Squared Error, have not been reported. Furthermore, details of the training and testing groups have not been elaborated. Without assessing the performance of training and testing steps, it is difficult to accept the accuracy reports provided here. Kindly elaborate and update the results. 

We understand the reviewer’s concerns and have updated our methods section to elaborate on the EEG analysis and the neural network process more clearly. Furthermore, we have added a table to provide both percentage accuracy and mean absolute percentage error for each subject. Furthermore, we have now elaborated the details about the training and testing groups.

Reviewer 2 Report

Dear Authors

I have had the opportunity to review your recent paper, and I appreciate the efforts you have put into your research. However, I have some concerns regarding the small sample size used in your study.

While I understand that small sample sizes can be a limitation in certain research settings, I believe it is important that you provide more support and justification for your choice of sample size. To this end, the method section could be improved, by adding some recent EEG studies that have used similar sample sizes in different backgrounds, and have successfully addressed these concerns; please include the following references:

1. Di Flumeri et al. “Brain–Computer Interface-Based Adaptive Automation to Prevent Out-Of-The-Loop Phenomenon in Air Traffic Controllers Dealing With Highly Automated Systems”

2.  Gomez et al. “User Engagement Comparison between Advergames and Traditional Advertising Using EEG: Does the User’s Engagement Influence Purchase Intention?”

3. Perera et al. “Improving EEG-Based Driver Distraction Classification Using Brain Connectivity Estimators”

This would help to build a stronger case for the validity and reliability of your findings, and would provide readers with a more comprehensive understanding of the limitations of your study.

In addition, in the discussion section, please add the potential impact of the small sample size on the validity and generalizability of your findings.

Best regards

Author Response

We thank the reviewer for the appreciative words about our work. We also thank the reviewer for giving us an opportunity for improving our paper by suggesting references to justify our small sample size. As such we have incorporated the references in our methods section and have listed the sample size as a limitation in the discussions too. 

Reviewer 3 Report

My main concern with the research is that the subjects are instructed when to do the MI with a visual cue. So it is hard to separate the evoked potential from the actual intention. 

I mean that the MI event should be not instructed by an external cue to be sure you are measuring the actual MI and not an ERP.

I would suggest the authors to discuse this part and justify or even asume that they are classifying an ERP to a visual cue to perform a MI action. This changes a lot the possible conclusions of the paper, but to be true, at least 2 seconds after the visual cue should be neglected before considering that the event analyzed is a pure MI mental task and not a visual ERP.

Other things:

Please ellaborate EEG analysis part. It is too short to allow reproducibility of the experiments;

-Define the ADJUST algorithm

-Define the methodology for the time frequency analysis in lines 185 to 189. Are you doing A-R/R*100?

-What is the value you are averaging power? Amplitude per frequency band?

-Which is the base for the dB scale? Please detail the methodology to allow the experiments to be reproduced.

-Please make the LDA part more clear to exactly define which features (electrodes and frequency bands) are you using in the classifier and which epochs (periods of time used). The paragraph is not clear enough, although you are defining part of the features. It needs to be clear which are the number of features you are using for each sample in the classifier.

Correct the . in line 182 between tasks and nine

Author Response

We thank the reviewer for the insightful comments and giving us the opportunity to improve our paper. We have addressed the comments as follows:

1. My main concern with the research is that the subjects are instructed when to do the MI with a visual cue. So it is hard to separate the evoked potential from the actual intention. 

I mean that the MI event should be not instructed by an external cue to be sure you are measuring the actual MI and not an ERP.

I would suggest the authors to discuse this part and justify or even asume that they are classifying an ERP to a visual cue to perform a MI action. This changes a lot the possible conclusions of the paper, but to be true, at least 2 seconds after the visual cue should be neglected before considering that the event analyzed is a pure MI mental task and not a visual ERP.

The reviewer is correct in noticing that there is a visual component in our motor imagery paradigm. However, the visual cue is not just a start command to evoke a visual ERP. Instead it is an animation of the action to be imagined to help the subjects pace their imagination to the action shown on-screen. This does introduce an action observation component to our motor imagery exercise. But to make sure the subjects did perform motor imagery, weinstructed the subjects to imagine the action kinesthetically. Also the spatial distribution figure does shows a contralateral right side activation corresponding to the motor imagery of the left hand fingers, eliminating any possibility of visual evoked potentials being dominant. We have incorporated a paragraph about this action observation combined with motor imagery and its benefits as part of the discussion. 

2. Please ellaborate EEG analysis part. It is too short to allow reproducibility of the experiments;

-Define the ADJUST algorithm

-Define the methodology for the time frequency analysis in lines 185 to 189. Are you doing A-R/R*100?

-What is the value you are averaging power? Amplitude per frequency band?

-Which is the base for the dB scale? Please detail the methodology to allow the experiments to be reproduced.

-Please make the LDA part more clear to exactly define which features (electrodes and frequency bands) are you using in the classifier and which epochs (periods of time used). The paragraph is not clear enough, although you are defining part of the features. It needs to be clear which are the number of features you are using for each sample in the classifier.

We have edited and added more information in the methods section as per the suggestions by the reviewer to offer more clarity and help in reproducibility of the study.

3. Correct the . in line 182 between tasks and nine

The line has been corrected now. 

Round 2

Reviewer 1 Report

No further comments. 

Author Response

We thank the reviewer for giving us the pportunity to improve our work. 

Reviewer 2 Report

After the revisions made by the authors, i suggest to accept the article for publication in the present form

Author Response

(The authors gave the same response as above.)

Reviewer 3 Report

My main concern is still remnant (ERP vs ERD/ERS). Even though the subject is doing a kinesthetic MI it is influenced by the visual feedback and it is not a voluntary mental action but one instructed. The only way to solve the issue is not studying the transitory, i.e. neglecting at least 2s after the cue, or accepting the limitations of the study. I think that this should be addressed by adding a subsection called limitations of the study explaining that even though the subject is doing a kinesthetic MI it is not possible to separate that from any possible ERP due to the visual feedback, as neglecting the transitory would require a full new approach to the research. This is a problem of the experimental protocol used and it must be specified in the manuscript to be accepted. The research has its value, but it is limited by the experimental conditions and it must be stated in order to be presented as a rigorous study.

Author Response

We thank the reviewer again for taking the time to review our work and give suggestions to improve the paper. We have taken the reviewer's advise and have made a limitations section. We have included the limitation with our experimental design as such in lines 372-378:

"The major limitation of the study is the presence of event-related potential (ERP) components from the action observation that might interact with the ERD elicited by the kinesthetic motor imagery. The experimental design in the current study makes it impossible to separate the AO related ERP components from the MI related ERD. Future studies can have practice sessions with AO component to train the subjects on the pacing and timing of the performed kinesthetic MI and for the actual experiment the AO is turned off and subjects are made to perform MI without a visual cue. " 

We thank the reviewer again for giving us an opportunity to improve our work. We hope the changes we made will suffice.

Best Regards